# Health-Related Quality of Life in Adults with Cystic Fibrosis: Familial, Occupational, Social, and Mental Health Predictors

**DOI:** 10.3390/healthcare10071351

**Published:** 2022-07-21

**Authors:** Julien Ancel, Claire Launois, Jeanne-Marie Perotin, Bruno Ravoninjatovo, Pauline Mulette, Jean Hagenburg, Julie Malet, Muriel Griffon, Sophie Carré, François Lebargy, Gaëtan Deslée, Sandra Dury

**Affiliations:** 1Department of Respiratory Diseases, Reims University Hospital, 45, Rue de Cognacq-Jay, CEDEX, 51 092 Reims, France; claunois@chu-reims.fr (C.L.); jmperotin-collard@chu-reims.fr (J.-M.P.); bravoninjatovo@chu-reims.fr (B.R.); pmulette@chu-reims.fr (P.M.); jhagenburg@chu-reims.fr (J.H.); jmalet@chu-reims.fr (J.M.); mgriffon@chu-reims.fr (M.G.); scarre@chu-reims.fr (S.C.); flebargy@chu-reims.fr (F.L.); gdeslee@chu-reims.fr (G.D.); sdury@chu-reims.fr (S.D.); 2INSERM UMRS 1250, University of Reims Champagne-Ardenne, 51 100 Reims, France; 3EA7509 IRMAIC, University of Reims Champagne-Ardenne, 51 100 Reims, France

**Keywords:** cystic fibrosis, quality of life, anxiety, depression, CFQ-R

## Abstract

Background: Cystic Fibrosis (CF) adult patients experience daily physical symptoms and disabilities that may impact their quality of life and mental health. Methods: This prospective study aimed to evaluate the relative contribution of the familial, occupational, and social environment, besides that of the main physical and mental health factors, to the quality of life of CF adult patients using the Cystic Fibrosis Questionnaire-Revised (CFQ-R) in a multivariate model. Results: Fifty patients were analyzed (70% of men; median age of 25 years; median body mass index of 21 kg/m²; median FEV1 of 57%). Anxiety and depression scores were negatively associated with 9 of the 12 CFQ-R domains. When controlling for anxiety and depression, FEV1% and BMI were significant positive predictors of several domains of the CFQ-R. All the familial, occupational, and social components analyzed but one (professional training) were predictors of at least one domain of the CFQ-R. Conclusion: Anxiety and depression explained a greater proportion of the variance than physical variables (age, sex, BMI, FEV1%, and exacerbation in the last year) in CF HRQoL. Many familial, occupational, and social components were also specifically and independently predictors of some HRQoL domains. Their screening might help identifying CF patients eligible for specific interventions, focusing on the impaired QoL dimensions.

## 1. Introduction

In the last decades, the median age of survival has nearly doubled for Cystic Fibrosis (CF) patients, being now over 40 years [1]. With the increase in overall survival, chronic CF-related disorders have become common, including diabetes mellitus [2], bone disease [3], reduced fertility [4], anxiety, and depression [5]. The treatment burden for both pulmonary and extra pulmonary CF long-term complications is growing in importance and can impact health-related quality of life (HRQoL) [6].

HRQoL is broadly assessed by the Cystic Fibrosis Questionnaire-Revised (CFQ-R), a well-validated multi-dimensional questionnaire that has demonstrated both validity, sensitivity, and reliability [7]. The CFQ-R was approved by the U.S. Food and Drug Administration for use in clinical trials [8,9]. The main publications that used this questionnaire aimed to establish correlations between CF HRQoL and clinical, bacteriological, or tomodensitometric features [10,11,12]. A recent systematic review of the literature showed that FEV1% predicted and pulmonary exacerbations were the strongest predictors of HRQoL [13]. However, the impact of mental health variables was underappreciated in this review. Some studies demonstrated that anxiety and depression were significant negative predictors of HRQoL [14,15,16,17]. Interestingly, anxiety and depression explained a greater proportion of variance overall than physical health variables [15,17,18]. Other dimensions such as education levels, geographical disparities, physical activity, or employment were rarely considered [19,20,21,22].

Our study aimed to examine the relative contribution of the main physical variables, mental health variables (anxiety and depression), and personal dimensions, with a focus on familial, occupational, and social data, on CF adult patients’ HRQoL using the CFQ-R in a multivariate model.

## 2. Materials and Methods

### 2.1. Study Design

This monocentric study was prospectively conducted at the Department of Respiratory Diseases (University Hospital of Reims, France) between November 2016 and December 2019. Patients were included in the RINNOPARI study (Recherche et INNOvation en PAthologie Respiratoire Inflammatoire), an observational cohort of inflammatory chronic lung diseases. The protocol was registered on clinicaltrials.gov (NCT02924818) on 5 October 2016. All patients received detailed information about the methods used and gave their written consent, as previously described [23]. 

Inclusion criteria were CF patients at least 18 years of age. Exclusion criteria were previous or planned lung transplantation, patients requiring an urgent visit, or any ongoing or recent medical condition in the last 4 weeks, including pulmonary exacerbations.

### 2.2. Patient Records

Anonymized data were collected and registered on an electronic medical record. They included demographic (age, sex), clinical (BMI, exacerbation in the last year), and functional data (FEV1, % predicted). Familial (partner status, living with parents, having children), occupational (student/professional training, professional activity), and social (social assistance, psychological follow-up, hobbies, sports activities) variables were specifically recorded.

### 2.3. Hospital Anxiety and Depression Scale 

Anxiety and depression screening measures were scored using a French validated version of the Hospital Anxiety and Depression scale (HADS), a self-administered rating scale specifically designed for patients with physical illness [24]. It consists of 14 items (all rated from 0 to 3), seven of which are related to depression (D), and the remaining seven to anxiety (A), with cut-off points for severity based on a score range from 0 to 21 for each domain, in which higher scores indicate higher suspicion of anxiety or depression. For example, the anxiety domain was explored using statements such as “I feel tense or wound-up …”, “I can sit at ease and feel relaxed…”, or “I get sudden feelings of panic …”, while the depression component was assessed with specific statements such as ‘I still enjoy the things I used to enjoy …”, “I can laugh and see the funny side of things …”, or “I feel cheerful ….” [25,26].

### 2.4. Cystic Fibrosis Questionnaire-Revised

The CFQ-R consists of 50 items structured in 12 independent domains, themselves divided into 6 domains assessing general aspects of HRQoL, i.e., physical functioning, role limitations, vitality, health perceptions, emotional state, and social functioning, and 6 domains related to specific aspects of CF, i.e., body image, eating problems, treatment burden, weight problems, respiratory symptoms, and digestive symptoms. All the answers were reported on a 4-point scale. The score for each scale ranges from 0 to 100, the higher score representing better quality of life [7,27]. 

### 2.5. Statistical Analyses

Data are expressed as median and range for quantitative variables and as number and percentage for qualitative variables. For each multivariable linear regression model, the CFQ-R domain score was entered as the dependent variable, and sex, age, FEV1 (% predicted), Body Mass Index (BMI) as independent variables. According to the collinearity between anxiety and depression (r = 0.556, *p* < 0.0001), the regression models had to be treated separately. As a consequence, the respective contributions of anxiety and depression could not be interpreted separately. Standardized regression coefficients (ß) were used as a measure of relative effect size, and squared part correlational coefficients were used to estimate the proportion of variance explained by each predictor in the model. *p*-values lower than 0.05 were considered statistically significant. XLSTAT software (version 2021.3.1, Addinsoft company, Paris, France) was used to analyze and reformat the data.

## 3. Results

Fifty-one consecutive CF patients were included in the study. One patient was excluded because of previous lung transplantation. Fifty patients were analyzed. No patient exited the study.

### 3.1. Patient Characteristics

Patients’ demographic, clinical, bacteriological, and functional characteristics are detailed in Table 1. Seventy percent of patients were men. The median age was 25 years, and the median BMI was 21 kg/m². The median FEV1 was 57% of the predicted value. Eight patients declared a history of depression, while none of them used anti-depressant. On HADS, the median anxiety and depression scores were 6 (3–8) and 3 (1–6), respectively. Linear regression revealed a significant correlation between anxiety and depression HADS (r = 0.556, *p* < 0.0001).

The familial, occupational, and social characteristics are detailed in Table 2. Thirty-three percent of patients lived with their parents, and 14% lived alone. Sixty-seven percent of the patients declared living with a partner, 6 of them living with their partner at the parental home. Fifty-one percent of the patients were employed, and 29% were students or were following a professional training program. Most patients had interests, including a sporting activity (74%) and/or hobbies (84%).

### 3.2. Quality of Life Scale 

The detailed results of the CFQ-R are shown in Table 3. “Vitality” was the most impaired domain, with a score of 58 (42–75), and “eating disturbance” was the least impaired domain, with a score of 100 (89–100).

### 3.3. Predictors of HRQOL

The proportion of the variance accounted for HADS anxiety and HADS depression is presented in Table 4. If significant, values for the regression model are represented for each predictor in multivariate analysis with HADS anxiety (Table 5) and HADS depression (Table 6) separately.

Anxiety was a significant negative predictor for 9 of the 12 CFQ-R domains and reached a higher significance for “emotion” (r = −0.75, *p* < 0.001). Anxiety was the only explicative variable for the “role perception” CFQ-R domain. The best explicating models based on the R^2^ coefficient were observed for the “emotion”, “health perception”, “vitality”, and “physical functioning” CFQ-R domains (R^2^= 0.71, 0.70, 0.56, and 0.55, respectively). In the regression models including anxiety as a predictor, age and exacerbation in the last year were significant negative predictors for 4 and 2 of the 12 domains, respectively. Both BMI and FEV1% were significant positive predictors for 2 of the 12 domains: a higher BMI was associated with better “body image” and “weight” self-perceptions (r = 0.35, *p* < 0.05 and r = 0.30, *p* < 0.05, respectively). Gender was a significant predictor for 3 of the 12 domains: being male was associated with higher CFQ-R scores in “physical functioning” and “vitality” but with lower scores in “weight”. 

Regarding the personal dimensions, professional activity, social assistance, psychological follow-up, and sports activities were significant positive predictors for 3, 1, 1, and 1 of the 12 domains, respectively. Partner presence, living with parents, having children, and hobbies were significant negative predictors for 1 of the 12 domains. Of note, the practice of sports significantly improved “vitality perception” (r = 0.24, *p* < 0.05). Social assistance was the only significant explicative variable associated with a worse “social perception” (r = −0.36, *p* < 0.05). Excluding the HADS-anxiety component, a higher “treatment burden” perception was only associated with professional activity (r = 0.37, *p* < 0.05).

Professional training was not a significant predictor for any of the CFQ-R domains. Interestingly, no significant association was identified for the “eating disturbance” CFQ-R domain. All values of the regression model for each predictor in multivariate analysis with HADS anxiety are summarized in Table 5.

The values for the regression model and the proportion of variance for each predictor of HADS depression are presented in Table 5. The best explicating models based on the R^2^ coefficient were observed for the “physical functioning”, “vitality”, “health perception”, “emotion”, and “body image” CFQ-R domains (R^2^ = 0.65, 0.65, 0.64, 0.61, and 0.57, respectively) (Table 4).

Depression was a significant negative predictor of 9 of the 12 CFQ-R domains. In the regression models in which depression was included as a predictor, BMI and FEV1% were significant positive predictors for 2 and 1 of the 12 domains, respectively. Gender was a significant predictor for 2 of the 12 domains: being male was associated with higher CFQ-R scores in “vitality” but with lower scores in “weight” domains. Age and exacerbation in the last year were not significant predictors for any of the CFQ-R domains’ scores. All values for the regression model for each predictor in multivariate analysis with HADS depression are summarized in Table 6.

Regarding the personal dimensions, professional training, professional activity, and psychological follow-up were each significant positive predictors for 1 of the 12 domains. Living with parents, having children, and social assistance were significant negative predictors for 2, 1, and 2 of the 12 domains, respectively. Partner presence was a significant predictor for 2 of the 12 domains: having a partner was associated with higher CFQ-R scores in the “emotion” dimension but with lower scores in the “digestive symptoms” dimension. Hobbies and sports were not significant predictors for any of the CFQ-R domains. Interestingly, no significant association was noticed for the “eating disturbance” CFQ-R domain.

## 4. Discussion

Our study aimed to assess the determinants affecting quality of life in CF adult patients using the CFQ-R. We chose to focus on the familial, occupational, and social components which are rarely considered and highlighted their impact on patients’ quality of life. 

As previously reported, we showed strong relationships between anxiety and depression and many domains of quality of life [14,15]. Anxiety and depression features separately impacted at a high degree the variance of the emotional domain. These results suggest that the presence of anxiety or depression were likely to have a large negative impact on the “emotion” HRQOL dimension. Both anxiety and depression are the main psychological features observed in CF patients and their caregivers [28]. Current guidelines recommend an annual screening of these psychological disorders [29]. However, the evaluation of anxiety and/or depression is insufficient to predict CF HRQoL, and additional predictors need to be established [30]. 

Our results are also consistent with previous reports investigating the most commonly studied variables in HRQoL domains, including gender, age, BMI, FEV1% predicted, and pulmonary exacerbation [31,32]. BMI was positively correlated with the “body image” and “weight” domains. Age was negatively correlated with the “physical functioning”, “vitality”, and “health perception” domains. We can hypothesize that the occurrence of CF-related comorbidities overtime impact self-perception HRQoL [33]. However, age was not negatively correlated with the “treatment burden” domain in our study. Pulmonary exacerbation in the past year was negatively associated with the “respiratory symptom” domain [34,35,36]. Male subjects harbored a better “physical functioning” domain than female subjects, which may be consistent with better survival and lung function in male CF subjects compared to female CF patients [37]. CF patients with higher predicted FEV1% also scored better in the “physical functioning” and “health perception” domains [17,38].

Our study focused on the relationships between familial, occupational, and social dimensions and CF HRQoL domains. First, we did find almost no association between current student/professional training and quality-of-life self-perception. However, working CF patients described a higher “treatment burden”, as previously reported [39]. In contrast, Havermans et al. did not find any relation between work status and treatment burden but with some specific domains of quality of life related to physical and psychosocial well-being [40]. Many adolescents and young adults did not report that work interfered with their daily CF therapies, though most of them disclosed their lung condition [41]. 

Very few studies investigated the impact of family status on CF HRQoL. Previous reports broadly focused on caregiver burden and quality of life of parents with CF children [42]. We highlighted that living with parents was negatively associated with the “weight” domain, and partner presence with the “digestive symptoms” domain. In a small study, CF patients and their partners seemed to be globally satisfied with their marital relationship and exhibited similar symptoms of anxiety and depression whatever the disease severity [43]. In another larger study, partner presence was a stronger predictor of life satisfaction and appeared to be helpful to maintain a good quality of life [44]. Regarding CF patients with children, we report a negative correlation between having children and the “emotion” HRQoL dimension and a similar trend for “treatment burden” or “health perception”. However, parenthood was only reported by four (8%) CF patients, potentially due to low fertility in patients with CF. This limited the extrapolation of our interpretation on this point. That is consistent with a recent survey noting that CF influenced women’s decision to have a child. For those becoming parents, greater satisfaction was counterbalanced with more concern about their health [45]. Such consideration is well reflected by the formulation “being a parent on compressed time”, illustrating the difficulty of raising children for parents with a limited life expectancy and subjected to daily treatment [46,47]. Of note, over time women described successful adaptation to managing parental commitments and CF [46,47].

In our study, sports practice was associated with an enhancement of “vitality” HRQoL. Most previous reports described no association between HRQoL domains and sports, with a limitation due to small numbers of patients and interventional study design [21]. Lastly, patients with social assistance were negatively impacted in the “social perception” domain. This is consistent with a previous review reporting that decreased physical and mental aspects of health correlated with decreased “social functioning” scores and that social support was an important aspect of CF adult life [48]. Identifying a negative social perception can help to propose personalized interventions [5,49].

There are some limitations to our study. Our study was monocentric, with a relatively small sample size, while other cohorts exploring new features in HRQoL remained equally dimensioned. We did not consider potential co-factors such as counselling and treatment for anxiety and depression, which represents a potential bias. We also used HADS as a surrogate to estimate anxiety and/or depression, while HADS was designed to screen psychological distress before admission to a hospital setting. Anxiety and depression as psychopathological categories (as labelled in the Diagnostic and Statistical Manual of Mental Disorders–DSM-5) should be assessed by specific tests in a further study. The transversal analysis could also be enhanced by a longitudinal cohort subjected to a specific intervention trying to improve the dedicated HRQoL domain. To note, lower R2 values reflect a poorer relevant model to predict accurate HRQoL with the evaluated parameters. For these conditions, low R2 values may limit the strength of the interpretations related to exploratory variables. Finally, our study was conducted before the large use of CFTR modulators such as elexacaftor/tezecaftor/ivacaftor, which are associated with a dramatic improvement in respiratory-related quality of life [50,51].

## 5. Conclusions

CF HRQoL is a complex construct resulting from both physical and psychological factors, sometimes inter-related. As previously reported, our study showed that anxiety and depression were strongly associated with CF HRQoL. Health mental features explained a higher proportion of the overall variance than the usual physical health parameters but remained insufficient to explain HRQoL. Our study highlights the importance of some familial, occupational, and social dimensions. These features were even specific and independent predictors for some HRQoL domains. In current CF management, their consideration might help to identify CF patients eligible for a specific intervention, focusing on the impaired QoL dimension.

## Figures and Tables

**Table 1 healthcare-10-01351-t001:** Demographic, clinical, functional, and microbiological characteristics.

Variables	*n* = 50
Male	35 (70)
Age, years	25 (21–32)
BMI, kg/m²	21 (20–23)
CFTR mutation	
ΔF508 homozygous	20 (40)
ΔF508 heterozygous	25 (50)
Other	5 (10)
Pancreatic insufficiency	40 (80)
Diabetes	17 (34)
Exacerbation in the last year	34 (68)
Colonization	
*Pseudomonas aeruginosa*	18 (36)
*Staphylococcus aureus*	31 (62)
FEV1, % predicted	57 (37–90)
HADS-Anxiety	6 (3–8)
HADS-Depression	3 (1–6)

Data are expressed as median [1st–3rd quartiles] or frequency (percentage). BMI: Body Mass Index, FEV1: Forced Expiratory Volume in the first second, HADS: Hospital Anxiety and Depression scale.

**Table 2 healthcare-10-01351-t002:** Familial, occupational, and personal characteristics.

Variables	*n* = 50
Familial status	
Partner	33 (67)
Living with their parents	16 (33)
Having children	4 (8)
Occupational status	
Student/Professional training	14 (29)
Professional activity	25 (51)
Social dimension	
Sport	36 (74)
Hobbies	41 (84)
Psychological follow-up	6 (13)
Social assistance	23 (47)

Data are expressed as frequency (percentage). One response was missing for each domain for one different patient alternatively.

**Table 3 healthcare-10-01351-t003:** Quality of life scale according to the CFQ-R.

CFQ-R Domain	*n* = 50
Physical functioning	75 (54–92)
Role perception	83 (67–94)
Vitality	58 (42–75)
Emotion	80 (60–98)
Social perception	61 (51–67)
Body image	67 (56–83)
Eating disturbance	100 (89–100)
Treatment burden	67 (44–72)
Health perception	67 (44–72)
Weight	67 (33–100)
Respiratory symptoms	71 (62–76)
Digestive symptoms	89 (78–100)

Data are expressed as median (1st–3rd quartiles). CFQ-R: Cystic Fibrosis questionnaire-revised.

**Table 4 healthcare-10-01351-t004:** Multiple regression analyses for the Cystic Fibrosis Questionnaire-Revised (CFQ-R) in CF adults for HADS anxiety and depression.

	Regression Model Adjusted R²			
Dependent Variable	HADS Anxiety Component		HADS Depression Component	
CFQ-R Domains	%	*p*-Value	%	*p*-Value
*Physical functioning*	55.3	**<0.0001**	**65.2**	**<0.0001**
*Role perception*	26.6	**0.001**	21.1	**0.003**
*Vitality*	**55.8**	**<0.0001**	**65.5**	**<0.0001**
*Emotion*	**70.6**	**<0.0001**	61.4	**<0.0001**
*Social perception*	16.3	**0.022**	23.6	**0.003**
*Body image*	46.6	**<0.0001**	57.1	**<0.0001**
*Eating disturbance*	7.6	ns	7.6	ns
*Treatment burden*	26.1	**0.008**	12.0	**0.023**
*Health perception*	**70.2**	**<0.0001**	**64.0**	**<0.0001**
*Weight*	31.9	**0.003**	31.9	**0.003**
*Respiratory symptoms*	22.7	**0.014**	32.1	**0.002**
*Digestive symptoms*	33.6	**0.001**	24.0	**0.011**

R^2^: Regression coefficient model. HADS: Hospital Anxiety and Depression scale. R^2^ bolded values represent the 3 higher values for each HADS component, ns: not significant.

**Table 5 healthcare-10-01351-t005:** Multiple regression analyses for the Cystic Fibrosis Questionnaire-Revised (CFQ-R) in CF adults for HADS anxiety.

Dependent Variable	Independent Variable	Standardized Regression Coefficient	
CFQ-R Domains	Predictors	β	*p*-Value
*Physical functioning*			
	Age	−0.328	**0.012**
	Sex–Male	0.500	**0.000**
	FEV1%	0.262	**0.032**
	HADS *anxiety*	−0.338	**0.005**
	Sport	0.211	0.068
*Role perception*			
	HADS *anxiety*	−0.516	**0.001**
*Vitality*			
	Age	−0.296	**0.010**
	Sex–Male	0.373	**0.002**
	HADS *anxiety*	−0.479	**<0.0001**
	Sport	0.242	**0.027**
*Emotion*			
	Age	−0.266	**0.009**
	HADS *anxiety*	−0.737	**<0.0001**
	Partner presence	0.155	0.088
	Having children	−0.192	**0.043**
	Professional activity	0.247	**0.014**
*Social perception*			
	Exacerbation (last year)	0.248	0.086
	Social assistance	0.356	**0.015**
*Body image*			
	BMI	0.299	**0.011**
	HADS *anxiety*	−0.545	**<0.0001**
	Hobbies	−0.217	0.059
*Treatment burden*			
	HADS *anxiety*	−0.343	**0.021**
	Having children	−0.272	0.072
	Professional activity	0.369	**0.014**
*Health perception*			
	Age	−0.342	**0.002**
	FEV1%	0.360	**0.001**
	Exacerbation (last year)	−0.202	**0.034**
	HADS *anxiety*	−0.480	**<0.0001**
	Having children	−0.179	0.070
	Professional activity	0.311	**0.003**
*Weight*			
	Sex–Male	−0.301	**0.027**
	BMI	0.352	**0.011**
	Living with their parents	−0.298	**0.033**
	Psychological follow-up	0.339	**0.018**
*Respiratory symptoms*			
	Exacerbation (last year)	−0.326	**0.041**
	HADS *anxiety*	−0.323	**0.043**
*Digestive symptoms*			
	HADS *anxiety*	−0.420	**0.002**
	Partner presence	−0.322	**0.017**
	Hobbies	−0.280	**0.036**

β: Coefficient of regression. BMI: Body Mass Index, FEV1: Forced Expiratory Volume in the first second (% predicted), HADS: Hospital Anxiety and Depression scale.

**Table 6 healthcare-10-01351-t006:** Multiple regression analyses for the Cystic Fibrosis Questionnaire-Revised (CFQ-R) in CF adults for HADS depression.

Dependent Variable	Independent Variable	Standardized Regression Coefficient	
CFQ-R Domains	Predictors	β	*p*-Value
*Physical functioning*			
	HADS *depression*	−0.505	**<0.0001**
*Role perception*			
	HAD *depression*	−0.459	**0.003**
*Vitality*			
	Sex–Male	0.358	**0.001**
	Exacerbation (last year)	−0.191	0.055
	HADS *depression*	−0.638	**<0.0001**
	Living with their parents	−0.230	0.051
	Professional training	0.337	**0.007**
*Emotion*			
	HADS *depression*	−0.707	**<0.0001**
	Partner presence	0.241	**0.021**
	Having children	−0.198	0.058
	Social assistance	−0.227	**0.028**
*Social perception*			
	HADS *depression*	−0.422	**0.003**
	Living with their parents	−0.277	**0.044**
*Body image*			
	BMI	0.393	**0.000**
	HADS *depression*	−0.592	**<0.0001**
	Social assistance	−0.252	**0.018**
*Treatment burden*			
	Professional activity	0.346	**0.023**
*Health perception*			
	FEV1 %	0.489	**<0.0001**
	HADS *depression*	−0.633	**<0.0001**
	Having children	−0.213	**0.031**
*Weight*			
	Sex–Male	−0.301	**0.027**
	BMI	0.352	**0.011**
	Living with their parents	−0.298	**0.033**
	Psychological follow-up	0.339	**0.018**
*Respiratory symptoms*			
	Exacerbation (last year)	−0.275	0.068
	HADS *depression*	−0.451	**0.004**
*Digestive symptoms*			
	HADS *depression*	−0.286	**0.046**
	Partner presence	−0.285	**0.046**
	Hobbies	−0.260	0.069

β: Coefficient of regression. BMI: Body Mass Index, FEV1: Forced Expiratory Volume in the first second (% predicted), HADS: Hospital Anxiety and Depression scale.

## Data Availability

Not applicable.

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
