# Peer review of "Health-Related Quality of Life in Adults with Cystic Fibrosis: Familial, Occupational, Social, and Mental Health Predictors"

_healthcare, 2022, doi:10.3390/healthcare10071351_

Round 1

Reviewer 1 Report

Introduction

1)      The introduction is well written, setting the context and explaining the rationale for this work.

Methods

2)      Since the patients were included in the RINNOPARI trial, the trial registration number and maybe a website or a reference to the trial protocol should be included.

3)      The hospital anxiety and depression scales appear briefly explained in the methods section. I would suggest explaining further the scoring system by mentioning the items and the range (maximum-minimum scores and their meaning) as the authors did with the CFQ-R scores. Do the emotion domain of the CFQ-R and the HADS-anxiety or HADS-depression overlap in some questions?

Results

4)      In Table 1, The median FEV1 % predicted is 57%. However, the Q1 and Q3 appear respectively as 20 and 23%. Please, double-check these values.

5)      The main concern with this paper is how data is presented. Tables 4 and 5 are certainly long, displaying lots of empty cells (and empty space) and showing almost all the “raw outcome” of all multivariate analyses. I would strongly suggest condensing the main outcomes of the study to improve the readability of the results.

6)      The R2 of many variables of the CFQ-R domain, although significant, shows modest R2 values. For instance, in HADS Anxiety, “role perception” R2 is 26.6; “social perception” R2 is 16.3, “Treatment burden” R2 is 26.1; “respiratory symptoms” R2 is 22.7. Low R2 values may limit the strength of some of the observations described by the authors. One strategy to present the results could be to present first the R2 values for all domains in a separate table and deep dive into those CFQ-R variables showing the highest and more meaningful R2.

7)      The CFQ-R variable “Emotion” achieved high adjusted R2 values (with p<0.0001), and the highest Beta coefficients for HADS-anxiety and HADS-depression were also found for the variable “emotion”. How do the authors interpret these results?

8)      How would the outcome of the analyses be impacted if both HADS-anxiety and HADS-depression were included in the same multivariate regression? If a single analysis including both HADS is feasible, it would also facilitate data presentation and conclusions.

Discussion

9)      L225. The authors state, “Regarding CF patients with children, we reported a negative correlation with the emotion dimension and similar trend for treatment burden or health perception.

However, the significance (significant or not) of the negative correlation with emotion varies depending on the multivariate analysis. Moreover, there are only 4 patients with children (out of 50 total patients, 8%). Do the authors consider this number enough to draw such a conclusion?

Minor

              L114, replace HAD by HADS for consistency.

Author Response

Dear editor, Dear Reviewer,

Thank you for your comments, which were of great help in the process of revision of the manuscript healthcare-1807453 entitled ‘Health-related quality of life in adults with cystic fibrosis: family, occupational, social, and mental health predictors’.

Below you will find a list of changes related to the manuscript. We have provided point-by- point responses. The changes made in the manuscript are highlighted with track changes. Thus, all changes can be read in the manuscript as well. We used the attached document of manuscript, as recommended.

Your sincerely,

On behalf of the co-authors

ANCEL Julien M.D.

Response to reviewer #1:

Introduction

The introduction is well written, setting the context and explaining the rationale for this work.

RESPONSE: We thank the reviewer for this comment.

Methods

Since the patients were included in the RINNOPARI trial, the trial registration number and maybe a website or a reference to the trial protocol should be included.

RESPONSE: Trial registration number was already presented in the section “Institutional Review Board Statement”. To highlight this point, we also reported the registration number in lines 65-66: “The protocol was registered on clinicaltrials.gov (NCT02924818) on 5th October 2016.”

The hospital anxiety and depression scales appear briefly explained in the methods section. I would suggest explaining further the scoring system by mentioning the items and the range (maximum-minimum scores and their meaning) as the authors did with the CFQ-R scores. Do the emotion domain of the CFQ-R and the HADS-anxiety or HADS-depression overlap in some questions?

RESPONSE: According with reviewer’s comment, we modified the material and method section as follow:

Line 82-83: It consists of 14 items « (all rated from 0 to 3) » has been added

Lines 84 to 89 : “: Score range from 0 to 21 for each domain, indicating high anxiety or depression component disorder for higher scores” has been added and “For example, Anxiety domain was explored by question as “I feel tense or wound-up …”, “I can si at ease and feel relaxed…” or “I get sudden feelings of panic …” while Depression component was assessed by specific items such as ‘I still enjoy the things I used to enjoy …”, “I can laugh and see the funny side of things …” or “I feel cheerful ….””

If necessary, complete HAD questionary form can be found at the reference 24 as specified.

Results

In Table 1, The median FEV1 % predicted is 57%. However, the Q1 and Q3 appear respectively as 20 and 23%. Please, double-check these values.

RESPONSE: Thanks to the reviewer for highlighting this mistake.

We corrected these values as follows in table 1: FEV1, % predicted: 57% “[37-90]”, data expressed as median [1st-3rd quartiles].

The main concern with this paper is how data is presented. Tables 4 and 5 are certainly long, displaying lots of empty cells (and empty space) and showing almost all the “raw outcome” of all multivariate analyses. I would strongly suggest condensing the main outcomes of the study to improve the readability of the results.

The R2 of many variables of the CFQ-R domain, although significant, shows modest R2 values. For instance, in HADS Anxiety, “role perception” R2 is 26.6; “social perception” R2 is 16.3, “Treatment burden” R2 is 26.1; “respiratory symptoms” R2 is 22.7. Low R2 values may limit the strength of some of the observations described by the authors. One strategy to present the results could be to present first the R2 values for all domains in a separate table and deep dive into those CFQ-R variables showing the highest and more meaningful R2.

RESPONSE: Comments 5 & 6 were treated together. According with this helpful comment, we deeply modified previous tables 4 & 5.

We thus introduced a new table 4 presenting R2 values and their respective p-value for both HADS- anxiety and depression as proposed by the reviewer in lines 152 to 156 with the following sentences: “The proportion of the variance accounted for HADS anxiety and HADS depression are presented in Table 4. If significant, values for the regression model are represented for each predictor in multivariate analysis with HADS anxiety (table 5) and HADS depression (table 6), separately.”

We then deleted all empty lines of previous tables 4 & 5, becoming current tables 5 & 6. Additionally, previous sentences “Best explicating models based on the R2 coefficient were observed for “emotion”, “health perception”, “vitality”, and “physical functioning” CFQ-R domains (R2= 0.71, 0.70, 0.56, and 0.55, respectively).” were moved at the beginning of the result section (in lines 218 to 220 and 309 to 311). Finally, we added in discussion section (lines 681 to 683) the mention: “To note, lower R2 values reflect poorer relevant model to predict accurate QoL with evaluated parameters. For these conditions, low R2 values may limit the strength of the interpretation related to exploratory variables”.

The CFQ-R variable “Emotion” achieved high adjusted R2 values (with p<0.0001), and the highest Beta coefficients for HADS-anxiety and HADS-depression were also found for the variable “emotion”. How do the authors interpret these results? 

RESPONSE:

We thank the reviewer for highlighting this interesting aspect of our results. As intuitive, emotion components are highly related with anxiety and depression features. This is well reported by Cronly J et al. Respir Care 2019. Moreover, the CFQ-R variable “Emotion” was the variable with highest beta coefficient and mainly due to both HADS anxiety and HADS depression. This result could also partly due to anxiety and depression collinearity and remain concordant with previous report (https://doi.org/10.4187/respcare.06356).

We thus introduced in the discussion the following sentence (lines 222 to 225): « Anxiety and depression features separately impacted at a high degree the variance the emotional domain. These results suggested that presence of anxiety or depression were likely to have a large negative impact on emotion HRQOL dimension. »

How would the outcome of the analyses be impacted if both HADS-anxiety and HADS-depression were included in the same multivariate regression? If a single analysis including both HADS is feasible, it would also facilitate data presentation and conclusions. 

RESPONSE:

We tried to explain this point more accurately in the paper.

As prespecified in method section, “anxiety and depression were entered into different models according to their collinearity (r=0.556, p<0.0001)”.  According with reviewer’s comment, we precised the methodological aspect in line 104 to 109: “According with collinearity between anxiety and depression (r=0.556, p<0.0001), regression models had to be treated separately. In consequence, respective contribution of anxiety and depression could not be interpreted from each other.”

Additionally, Anxiety and Depression are two components with high interrelationship and collinearity. Their separate analysis is common approach previously reported. We provided here some previous publication with this similar approach:

http://dx.doi.org/10.1016/j.genhosppsych.2016.02.002; https://doi.org/10.4187/respcare.06356  

Discussion

9)      L225. The authors state, “Regarding CF patients with children, we reported a negative correlation with the emotion dimension and similar trend for treatment burden or health perception. 

However, the significance (significant or not) of the negative correlation with emotion varies depending on the multivariate analysis. Moreover, there are only 4 patients with children (out of 50 total patients, 8%). Do the authors consider this number enough to draw such a conclusion?

RESPONSE:

This a relevant point discussed by the reviewer. We consent that only few patients (n=4 – 8%) had children and it must limit interpretation about these results. However, parenthood is a very rare aspect in CF disease due to lower fertility and our results could bring interesting data in this field. Moreover, children presence seemed to impact emotion dimension in multivariate analysis, despite strong significance related to anxiety / depression component.

With the agree of the reviewer, we proposed to introduce a limitation sentence to counterbalance our conclusion in lines 608 to 610:” However, parenthood was only reported in 4 (8%) of CF patients, potentially due to lower fertility in CF disease. This restricted effective must limit the extrapolation of our interpretation on this point.”. If reviewer estimate that this point is too critical, we could also withdraw these interpretations.

Minor

              L114, replace HAD by HADS for consistency.

RESPONSE:

We checked the whole manuscript and replaced HAD by HADS for consistency.

We hope we have sufficiently rebutted the comments of the reviewers and that these changes and responses will facilitate the decision to publish this manuscript in Healthcare. In any case, we would be glad to respond to any further questions and comments the reviewers may have.

Reviewer 2 Report

Cystic fibrosis is an important clinical and social problem. The modern development of medicine has allowed to increase the duration and quality of life of these patients. The study presented in this article is interesting and clinically relevant. The article contains an analysis of factors affecting the quality of life of adult patients with cystic fibrosis. However, there are a few comments.

Comments:

1.      There is no indication in the materials and methods as to whether any of the patients had received counseling and treatment for anxiety and depression or other mental health problems previously, including at the time of the study. This information is relevant.

2.      Were all patients questionnaires administered under the same conditions (out of the exacerbation period, against adequate treatment, etc.)? It is recommended to add information about this.

3.      Did all patients answer the questionnaires in their native language? Or were validated language versions used?

Author Response

Dear editor, Dear Reviewer,

Thank you for your comments, which were of great help in the process of revision of the manuscript healthcare-1807453 entitled ‘Health-related quality of life in adults with cystic fibrosis: family, occupational, social, and mental health predictors’.

Below you will find a list of changes related to the manuscript. We have provided point-by- point responses. The changes made in the manuscript are highlighted with track changes. Thus, all changes can be read in the manuscript as well. We used the attached document of manuscript, as recommended.

Your sincerely,

On behalf of the co-authors

ANCEL Julien M.D.

Response to reviewer #2:

Cystic fibrosis is an important clinical and social problem. The modern development of medicine has allowed to increase the duration and quality of life of these patients. The study presented in this article is interesting and clinically relevant. The article contains an analysis of factors affecting the quality of life of adult patients with cystic fibrosis.

RESPONSE: We thank the reviewer for this remark.

Comments:

  1. There is no indication in the materials and methods as to whether any of the patients had received counselling and treatment for anxiety and depression or other mental health problems previously, including at the time of the study. This information is relevant.

RESPONSE:

Thanks to the reviewer, that was a critical point with high relevance. As a transversal study, we only collected declarative data, wondering “are there depression diagnosed in the past?” representing 8 patients. Unfortunately, we do not have data about counselling and anti-depressant treatment.

We thus added these data in line 122-123: “8 patients declared a past history of depression while any of them used anti-depressant”. Additionally, missing data were discussed in lines 628-629: “We did not collect potential co-factors such as counselling and treatment for anxiety and depression, representing potential bias”.

  1. Were all patients questionnaires administered under the same conditions (out of the exacerbation period, against adequate treatment, etc.)? It is recommended to add information about this.

RESPONSE:

We apologize about our lack or precision on this point. Inclusions were realized according with the RINNOPARI protocol. Exclusion criterion were an ongoing or recent (< 4 last weeks) medical condition, including pulmonary exacerbations. We added this point in lines 70-71: « or any ongoing or recent medical condition in the last 4 weeks, including pulmonary exacerbations. »

  1. Did all patients answer the questionnaires in their native language? Or were validated language versions used?

RESPONSE:

We thank again the reviewer for this valuable comment. For HADS questionnaires, a validated French version were used. We added the following sentence as follows:

Lines 80-81: « … using a French validated version of » and the following reference was added: [24] referencing:

Friedman, S.; Samuelian, J.C.; Lancrenon, S.; Even, C.; Chiarelli, P. Three-Dimensional Structure of the Hospital Anxiety and Depression Scale in a Large French Primary Care Population Suffering from Major Depression. Psychiatry Res 2001,104, 247–257, doi:10.1016/s0165-1781(01)00309-2.

For the CFQ-R, this tool was originally developed in French, and so we did not need to use a translated version. We added the adapted reference of the original publication for the CFQ-R in line 100 [27] referencing:

Henry, B.; Aussage, P.; Grosskopf, C.; Goehrs, J.-M. Development of the Cystic Fibrosis Questionnaire (CFQ) for Assessing Quality of Life in Pediatric and Adult Patients. Qual Life Res 2003, 12, 63–76, doi:10.1023/a:1022037320039.

We hope we have sufficiently rebutted the comments of the reviewers and that these changes and responses will facilitate the decision to publish this manuscript in Healthcare. In any case, we would be glad to respond to any further questions and comments the reviewers may have.

Round 2

Reviewer 1 Report

my comments have been addressed adequately

Author Response

Thanks again for your comments and your valuable time